# Anti-Inflammatory Effects of SGLT2 Inhibitors: Focus on Macrophages

**DOI:** 10.3390/ijms26041670

**Published:** 2025-02-15

**Authors:** Elena Y. Rykova, Vadim V. Klimontov, Elena Shmakova, Anton I. Korbut, Tatyana I. Merkulova, Julia Kzhyshkowska

**Affiliations:** 1Institute of Cytology and Genetics, Siberian Branch of Russian Academy of Sciences (IC&G SB RAS), Lavrentjev Prospect 10, 630090 Novosibirsk, Russia; rykova.elena.2014@gmail.com (E.Y.R.); klimontov@mail.ru (V.V.K.); kazakova.e.o@mail.ru (E.S.); korbutai@icgbio.ru (A.I.K.); merkulova@bionet.nsc.ru (T.I.M.); 2Research Institute of Clinical and Experimental Lymphology, Branch of the Institute of Cytology and Genetics, Siberian Branch of Russian Academy of Sciences (RICEL—Branch of IC&G SB RAS), Timakov Str. 2, 630060 Novosibirsk, Russia; 3Laboratory of Translational Cellular and Molecular Biomedicine, National Research Tomsk State University, 634050 Tomsk, Russia; 4Institute of Transfusion Medicine and Immunology, Institute for Innate Immunoscience (MI3), Medical Faculty Mannheim, University of Heidelberg, 68167 Mannheim, Germany

**Keywords:** diabetes, chronic kidney disease, heart failure, sodium–glucose cotransporter 2 inhibitor, inflammation, macrophages, signaling pathway

## Abstract

A growing body of evidence indicates that nonglycemic effects of sodium–glucose cotransporter 2 (SGLT2) inhibitors play an important role in the protective effects of these drugs in diabetes, chronic kidney disease, and heart failure. In recent years, the anti-inflammatory potential of SGLT2 inhibitors has been actively studied. This review summarizes results of clinical and experimental studies on the anti-inflammatory activity of SGLT2 inhibitors, with a special focus on their effects on macrophages, key drivers of metabolic inflammation. In patients with type 2 diabetes, therapy with SGLT2 inhibitors reduces levels of inflammatory mediators. In diabetic and non-diabetic animal models, SGLT2 inhibitors control low-grade inflammation by suppressing inflammatory activation of tissue macrophages, recruitment of monocytes from the bloodstream, and macrophage polarization towards the M1 phenotype. The molecular mechanisms of the effects of SGLT2 inhibitors on macrophages include an attenuation of inflammasome activity and inhibition of the TLR4/NF-κB pathway, as well as modulation of other signaling pathways (AMPK, PI3K/Akt, ERK 1/2-MAPK, and JAKs/STAT). The review discusses the state-of-the-art concepts and prospects of further investigations that are needed to obtain a deeper insight into the mechanisms underlying the effects of SGLT2 inhibitors on the molecular, cellular, and physiological levels.

## 1. Introduction

According to estimates from the International Diabetes Federation, the number of people with diabetes has reached 537 million worldwide in 2021. It is predicted that by 2045, 784 million people will be living with this disease [1]. Diabetes is one of the leading causes of mortality, blindness, end-stage renal disease, cardiovascular disease, heart failure, stroke, and lower limb amputations worldwide [2,3]. The modern diabetes management strategy involves individualization of treatment taking into account the range of complications and associated diseases [4]. Sodium–glucose cotransporter-2 (SGLT2) inhibitors are an essential element of cardiovascular and renal prevention in complicated diabetes, and their introduction into clinical practice opened up new opportunities for prevention and therapy of diabetic complications [2,5,6,7]. Accumulating evidence points to nonglycemic effects of these drugs which may be important for cardiovascular and renal protection [8,9,10,11]. In the DAPA-HF (Dapagliflozin and Prevention of Adverse Outcomes in Heart Failure), DAPA-CKD (Dapagliflozin and Prevention of Adverse Outcomes in Chronic Kidney Disease), and EMPEROR (Empagliflozin Outcome Trial in Patients with Chronic Heart Failure) trials, it was shown that SGLT2 inhibitors can decrease the risk of progression of heart failure and chronic kidney disease (CKD) in people with and without diabetes [11,12,13,14]. This confirms the standpoint that protective effects of SGLT2 inhibitors are realized not only through their antihyperglycemic activity.

Among the nonglycemic effects of SGLT2 inhibitors, the ability to control inflammation can be one of the main mechanisms underlying the protective effect on the target organs [15,16,17,18]. Chronic low-grade inflammation in the cardiovascular system, kidneys, adipose tissue, and other organs is a driver and consequence of diabetes [19,20,21]. Macrophages are the principal players in the initiation and maintenance of low-grade inflammation [22,23]. In vitro, macrophages can respond to cytokines and hormones to develop either proinflammatory (M1) or anti-inflammatory (M2) phenotypes. However, in vivo, such extreme variants of polarization are rather artificial, and macrophages can exhibit features of both M1 and M2 prototypes, as it has been recently identified in cancer [24]. Numerous studies indicate the involvement of inflammatory macrophages in the development of diabetic complications [25,26,27,28]. It was shown that SGLT2 inhibitors decrease the levels of proinflammatory cytokines in patients with type 2 diabetes, as well as the activity of NLRP3 inflammasome [29,30,31,32]. In agreement, some experimental studies have demonstrated the anti-inflammatory effect of SGLT2 inhibitors associated with attenuation of macrophage recruitment [16,33,34]. Anti-inflammatory effects of SGLT2 inhibitors in atherosclerosis, non-alcoholic fatty liver disease, chronic inflammatory bowel disease, Alzheimer’s disease, viral infections, and other diseases were also reported [11,16,35,36].

In this narrative review, we summarized the evidence for the anti-inflammatory effects of SGLT2 inhibitors with a particular focus on macrophages. Data from both clinical and experimental studies are discussed.

## 2. Macrophages in Diabetes-Related Inflammation

Chronic low-grade inflammation is involved in the pathogenesis of diabetes and diabetic complications, as well as in cardiovascular disease, non-alcoholic fatty liver disease, and cognitive dysfunction [20,21,23,37,38]. In this type of inflammation, both resident and monocyte-derived macrophages play an essential role [39]. Two major phenotypes, M1 and M2, are historically considered as major vectors of macrophage polarization [40,41,42,43]. However, the remarkable feature of macrophages is their ability to change phenotype depending on the simultaneous or sequential external signals [44,45]. Under physiological conditions, the resident tissue-specific M2 macrophages, which are frequently categorized as M2, are involved in maintenance of homeostasis and determine the stability of the microenvironment of organs and tissues. The M2 phenotype is induced by interleukin-4 (IL-4) and interleukin-13 (IL-13) which suppress an inflammatory response [42,44]. The M2 macrophages are characterized by high expression of CD206, CD163, and arginase 1 (Arg-1), and they secrete anti-inflammatory cytokines and extracellular matrix remodeling enzymes as well as structural ECM components that can lead to fibrosis [23,44,46,47,48].

Diabetes-related metabolic changes, such as hyperglycemia and accumulation of advanced glycation end products (AGEs) and oxidized lipoproteins, induce an inflammatory response in the cells of visceral fat, kidney, liver, myocardium, and vascular endothelium [21,49,50,51]. Hyperglycemia and dyslipidemia result in the elevated circulating levels of proinflammatory cytokines, such as tumor necrosis factor alpha (TNF-α), interleukin-6 (IL-6), interleukin-1β (IL-1β), and C-reactive protein (CRP) which, in turn, increase the risk of diabetes and vascular complications [21,52,53,54]. In addition, vascular endothelial cells at the site of inflammation secrete monocyte chemoattractant protein 1 (CCL2 or MCP-1) and express monocyte adhesion factors, such as vascular cell adhesion molecule 1 (VCAM-1), intercellular adhesion molecule 1 (ICAM-1), and platelet endothelial cell adhesion molecule 1 (PECAM-1) [39,55,56,57,58]. These factors attract resident macrophages and recruit monocytes from blood, promote their differentiation into inflammatory M1 phenotype, characterized by the enhance secretion of TNF-α, IL-1β, IL-6, and reactive oxygen species, which are involved in the injury of organs and tissues [23,39,59,60]. The inflammatory factors can cause metabolic reprogramming of the macrophages, switching them from oxidative to glycolytic metabolism [61,62,63]. However, metabolic factors themselves can stimulate inflammatory responses and proinflammatory programming in macrophages. Thus, hyperglycemia stimulates macrophages to produce proinflammatory factors macrophages (TNF-alpha, IL1-beta, S100A9, S10012), elevates expression of TLRs, and drives epigenetic changes supporting hyperglycemic memory related to vascular complications [25,64,65,66].

New data on the role of macrophage subtypes in the development of inflammation and tissue regeneration were obtained using single-cell techniques in a unilateral ureteric obstruction model [67]. It was demonstrated that monocytes recruited to the kidney early after injury rapidly adopt a proinflammatory and pro-fibrotic phenotype and express *Arg1*, and later transit to become *Ccr2^+^* macrophages that accumulate in late injury [67]. The authors found that a novel *Mmp12^+^* macrophage subset acts during the repair, which challenges the concept of M1/M2 dichotomy and demonstrates that functional phenotypes of macrophages are characterized by the most prominent functional biomarkers, as it was also recently demonstrated for macrophages in cancer [68,69,70,71]. scRNA-seq data on CD45+ immune cells obtained in diabetic OVE26 mice suggested that the macrophage polarization in diabetic kidney disease is a continuous process with an increased transition towards the M1 phenotype as well as with the appearance of numerous macrophage subsets at intermediate stages of their polarization and differentiation [27]. Figure 1 provides an overview for the major changes in the monocyte/macrophage system in diabetes.

## 3. Effects of SGLT2 Inhibitors on Inflammatory Markers in Patients with Diabetes

A number of clinical studies have documented the decrease in the levels of inflammatory markers in patients with type 2 diabetes treated with SGLT2 inhibitors (Table 1). In the EMPA-REG OUTCOME trial, a decrease in hs-CRP concentrations was found in patients with type 2 diabetes and high cardiovascular risk after a year of treatment with empagliflozin [72]. In the CANOSSA trial, canagliflozin lowered hs-CRP after 3, 6, and 12 months of therapy in subjects with type 2 diabetes and chronic heart failure [21,73]. A cohort of people with type 2 diabetes treated with empagliflozin for 24 weeks had decreased serum levels of hs-CRP and increased levels of anti-inflammatory IL-10 [74].

Subjects with type 2 diabetes treated with SGLT2 inhibitors demonstrated lower plasma IL-6 levels compared to the patients treated with other antihyperglycemic agents [30]. A reduction of IL-6 concentrations was found in the individuals with type 2 diabetes and heart failure after 3 months of treatment with empagliflozin [75]. The meta-analysis of randomized clinical trials comprising 18 studies and 5311 patients showed that dapagliflozin, empagliflozin, and canagliflozin considerably decrease IL-6 levels [29]. In the CANTATA-SU trial, the effect of canagliflozin was matched with that of glimepiride, a second-generation sulfonylurea. When compared with glimepiride, canagliflozin reduced plasma levels of IL-6, tumor necrosis factor receptor 1 (TNFR1), matrix metalloproteinase 7, and fibronectin 1, suggesting the inhibition of inflammation and fibrosis [76]. In patients with type 2 diabetes and a high cardiovascular risk, 30 days of therapy with empagliflozin reduced IL-1β and TNF-α production by human macrophages generated from PBMC ex vivo in presence of M-CSF; serum levels of IL-18 and IL-1β were decreased as well [31]. In the CANVAS study, canagliflozin decreased plasma concentrations of TNFR-1 and TNFR-2. After a year of treatment with canagliflozin, each 10%-step reduction of TNFR-1 and TNFR-2 concentrations was associated with a further reduction of risk for kidney complications [77].

The reported decrease in the levels of principal inflammatory markers in patients with type 2 diabetes treated with SGLT2 inhibitors is presented in Figure 2.

Recently, serum proteome and metabolome of patients with type 2 diabetes were assessed before and after dapagliflozin treatment with high-performance liquid chromatography followed by mass spectrometry. The therapy over 12 weeks affected not only glycolysis/gluconeogenesis and the pentose phosphate pathway, but also increased serum levels of disintegrin, metalloprotease-like decysin-1, and apolipoprotein A-IV (ADAMDEC1 and APOA4), the anti-inflammatory proteins mainly synthesized in the human gastrointestinal tract. The treatment was accompanied by a decrease in complement component 3 (C3), the mediator that promotes insulin resistance and inflammation [78].

Thus, the results of clinical studies clearly indicate the reduction of key markers of chronic low-grade inflammation in patients with T2D under treatment with SGLT2 inhibitors. This effect may be associated with a reduced risk of cardiovascular and renal complications.

The anti-inflammatory action of SGLT2 inhibitors raised hopes that these drugs would be useful in the treatment of COVID-19. However, clinical trials have not shown clinically significant effects [79]. It should be noted that SGLT2 inhibitors can hardly be considered anti-inflammatory drugs in the classical sense. Apparently, their anti-inflammatory effects are secondary to the metabolic action and are realized in metabolic-associated low-grade inflammation.

## 4. Anti-Inflammatory Effects of SGLT2 Inhibitors in Animal Models

A large set of studies has assessed the utility of SGLT2 inhibitors as potential anti-inflammatory therapeutics in a broad spectrum of animal models [16]. This was inspired by the positive effects of SGLT2 inhibitors in patients with metabolic and inflammatory pathologies, such as DM2 [80] and associated heart failure [81], diabetic cardiomyopathy and nephropathy [80], decrease in cognitive functions, non-alcoholic fatty liver disease (NAFLD) [82], hypertension [83], and other pathologies [11]. These studies demonstrated the effect of SGLT2 inhibitors on immune cells, preferentially macrophages, by suppressing their inflammatory activities. To decipher the cellular and molecular mechanism of action of SGLT2 inhibitors, animal models for type 2 diabetes, obesity, and cardiovascular diseases and other animal models of pathological inflammation were used by a number of research groups.

To induce severe atherosclerosis, wild-type C57BL/6J mice were treated with weekly injections of antisense oligonucleotides against the ldlr and srb1 mRNA in combination with a high cholesterol diet, followed by i.p. injections of streptozotocin to induce diabetes. In this type 2 diabetes model, the proliferation of resident macrophages in the plaques, as well as the leukocyte adhesion to the vessel wall, was considerably reduced under empagliflozin treatment [84]. The atherosclerotic plaques in these animals were considerably smaller, contained less lipids and CD68+ macrophages, and were enriched in collagen compared to mice without the therapy [84]. In a model of atherosclerosis induced by a high cholesterol diet followed by a balloon catheter injury of the abdominal aorta in New Zealand white rabbits, administration of dapagliflozin decreased the macrophage infiltration, reduced expression of proinflammatory cytokines (TNF-α, IL-1β, and IL-6) in the atherosclerotic plaques, and shifted macrophage polarization from M1 to M2 phenotype [85].

SGLT2 inhibitors effectively reduce key contributors to atherosclerosis, such as macrophage infiltration and pathological changes in the scavenging function, resulting in a foam cell formation [86]. The pattern of expression of scavenger receptors is defines the ability of macrophages to recognize, internalize, and degrade modified lipoproteins [87,88]. Empagliflozin specifically decreased macrophage infiltration and downregulated expression of scavenger receptors CD36 and LOX-1, impairing the ability of macrophages to absorb oxidized LDL and form foam cells in *db/db* mice [86]. In mouse diabetic models, treatment with empagliflozin and dapagliflozin significantly suppressed macrophage proliferation and leukocyte adhesion, leading to a notable reduction in plaque size [84,89].

In the model of hindlimb ischemia in C57BL/6 mice, dapagliflozin upregulated expression of angiogenic markers CD31 and α-SMA, and accelerated the blood flow in the affected limb phenotype [90]. In ischemic muscles, dapagliflozin induced the phenotypic shift of resident macrophages from M1 (iNOS+) towards M2 (Arg-1+) phenotype [90].

Arterial hypertension is also associated with low-grade inflammation in the cardiovascular system, kidneys, and lungs [91]. In spontaneously hypertensive rats (SHRs), empagliflozin decreased the inflammatory response and blood pressure [36]. This anti-inflammatory effect was examined in detail by the identification of the transcriptome (RNA-seq) of eight tissues (atrium, aorta, ventricle, white and brown adipose tissue, kidney, lung, and brain). In particular, the inflammation-associated MAPK10 expression in the kidneys was downregulated as well as the expression of the proinflammatory legumain and cathepsin S, while the expression of the anti-inflammatory AMP-activated protein kinases Prkaa1 and Prkaa2 was upregulated [36].

The neuroinflammation mediated by microglia (resident macrophages in the brain) is believed to play an important role in the cognitive impairment associated with diabetes [38]. In a murine model for a combination of Alzheimer’s disease and type 2 diabetes, empagliflozin decreased the vascular inflammation and the number of microglial cells in the brain [92]. These effects were accompanied by the improvement of cognitive functions [92]. In a model of diabetes induced by fructose and streptozotocin in CD-1 mice, SGLT2 inhibitors (empagliflozin and dapagliflozin) contributed to the multifactorial neuroprotection [38]. This effect included alleviation of cognitive impairment via restoring the levels of neurotrophins, modulation of neuroinflammatory signals with decrease in TNF-α, and expression of Snca, Bdnf, and App genes in the brain [38].

SGLT2 inhibitors can reduce the volume of adipose tissue, the size of adipocytes, and macrophage inflammatory activity in the adipose tissue [93,94]. In particular, ipragliflozin promoted a “healthy adipose tissue expansion” in the mice on high-fat diet. This effect was associated with a decrease in M1 and increase in M2 macrophages in the adipose tissue [95]. In mice fed with a high-fat diet, ipragliflozin reduced inflammation, fibrosis, and cell death in perirenal adipose tissue. Additionally, it promoted a shift in macrophage characteristics towards the anti-inflammatory M2 (CD11c-CD206+) profile [96]. The reduction of M1-polarized macrophage accumulation during the induction of anti-inflammatory M2 macrophage phenotype in the white adipose tissue and liver, decrease in the plasma TNF-α level, and alleviation of the obesity-associated inflammation was found under empagliflozin administration in high-fat diet-treated mice [97].

In a murine model of kidney fibrosis induced by the ischemia/reperfusion, a protective effect of dapagliflozin was demonstrated by the analysis of transcriptome and metabolome of the renal cortex. Dapagliflozin was shown to considerably mitigate the inflammation in fibrotic renal cortical tissue [32]. It restored the oxidation of fatty acids, preferable in tubular cells, and decreased the accumulation of tricarboxylic acid cycle metabolites. Oxidation of fatty acids is major energy source of M2 macrophages, in contrast to the M1 macrophages that use glycolysis [98]. Facilitating this metabolic pathway can have a potent and prolonged anti-inflammatory effect. The endogenous immunomodulatory metabolite itaconate, boosted by the effect of dapagliflozin, was shown to be involved in the inhibition of NLRP3 inflammasome activation [32]. In BALB/c mice, dapagliflozin alleviated a viral myocarditis induced by Coxsackie virus B3, increased the survival rate, and improved the heart function [34]. Dapagliflozin also decreased the levels of proinflammatory cytokines IL-1β, IL-6, and TNF-α, inhibited the macrophage differentiation towards M1 phenotype in the heart, and activated the STAT3 signaling pathway [34]. Canagliflozin decreased the M1/M2 ratio in the C57BL/6 mice with lipopolysaccharide (LPS)-induced lung injury and attenuated the inflammatory response [99]. A beneficial effect of empagliflozin on the immune response to influenza was reported [100]. Empagliflozin downregulated the IL-1β, IL-6, and CCL2 expression in the lungs of infected C57BL/6 mice and decreased the amount of proinflammatory monocytes with the CD45+CD11b+F4/80+LyC+ phenotype in the bronchoalveolar fluid. This agent was shown to inhibit the differentiation of proinflammatory M1 macrophages; however, the total number of CD45+CD11b−CD11c+SiglecF+CD64+ alveolar macrophages in the lungs, which preserve homeostasis, and the amount of proinflammatory CD206+ M2 macrophages was not affected [100].

Although the effect of SGLT2 inhibitors on intestinal inflammation in humans is vague, some positive effects have been observed in animal models. Specifically, SGLT2 inhibitors lowered the biochemical parameters of inflammation and the macro- and microinjuries of the colon in the acetic acid-induced colitis in rats [101,102,103]. In particular, empagliflozin decreased serum CRP and lactate dehydrogenase activity, and reduced expression of the inflammation mediators TNF-α, IL-1β, and IL-6 in the colon [104]. The effect of empagliflozin was analyzed in ulcerative colitis induced with dextran sulfate sodium (DSS) intoxication in rats. Administration of empagliflozin decreased an elevated level of inflammatory markers, including TNF-α, IL-1β, and IL-6, in colonic tissue. The agent downregulated the expression of tight junction proteins, including CLDN-2, CLDN-3, CLDN-4, CLDN-7, and CLDN-10. Analogously, the DSS-induced colitis in rats was alleviated by empagliflozin alone or in combination with metformin [105]. Interestingly, the combination of empagliflozin and metformin considerably enhanced AMPK phosphorylation and inhibited mTOR and NLRP3, leading to a subsequent decrease in caspase-1 cleavage and inhibition of a number of inflammatory cytokines, including IL-1β and IL-18 [105]. Moreover, empagliflozin has been shown to have renoprotective effects in chronic kidney disease (CKD). In a study using 5/6 nephrectomized rats with established CKD empagliflozin-attenuated renal fibrosis, empagliflozin prevented the polarization of pro-fibrotic CD68+CD206+ M2 macrophages by targeting mTOR and mitophagy pathways. Empagliflozin also attenuated inflammatory signals from CD8+ effector T-cells, which contribute to renal inflammation and fibrosis. By modulating these immune responses, empagliflozin was able to reduce renal fibrosis in the rat model of CKD. These findings suggest that empagliflozin’s renoprotective effects may be mediated, in part, by its ability to target immune pathways and to attenuate inflammation and fibrosis in the kidney [106].

Thus, the anti-inflammatory effects of SGLT2 inhibitors were observed in both diabetic and non-diabetic animal models (summarized in Table 2). The inhibition of the proinflammatory macrophage transformation plays an important role in the anti-inflammatory effect of SGLT2 inhibitors. The majority of studies in animal models found that the major and essential cellular mechanism controlled by SGLT2 inhibitors in vivo is the reprogramming of macrophages from the proinflammatory M1 to anti-inflammatory M2 phenotype. The molecular pathway changes in response to SGLT2 inhibitors can include modulation of scavenging function, changing metabolic pathways, activating anti-inflammatory signaling pathways, and suppression of inflammasome activity; however, which molecular interactions are primary and direct targets of SGLT2 inhibitors is still unclear.

## 5. Effects of SGLT2 Inhibitors on Macrophages In Vitro

Most studies that used cell lines have demonstrated a direct inhibitory effect of SGLT2 inhibitors on the production and secretion of proinflammatory cytokines in macrophages. SGLT2 inhibitors attenuated the activation of mouse RAW 264.7 macrophages induced by bacterial lipopolysaccharide (LPS) [100,103,111,112]. In several studies with the use of this line, it was demonstrated that SGLT2 inhibitors decrease the production of NO and IL-1α as well as the secretion of TNF-α, IL-1β, IL-6, IFN-γ, and proinflammatory chemokines CCL2, CCL3, CCL4, CCL5, and CXCL10. This decrease in the proinflammatory response was shown for canagliflozin [111], empagliflozin [100,103,112], and dapagliflozin [103]. The combination of empagliflozin and gemigliptin, an inhibitor of dipeptidylpeptidase-4, decreased TNF-α, IL-1β, IL-6, IFN-γ, CCL3, CCL4, CCL5, and CXCL10 expression and secretion in LPS-induced RAW 264.7 macrophages, demonstrating synergistic effects of both agents [112].

An anti-inflammatory effect of empagliflozin was also observed in a model of immune response to viral infection [100]. Empagliflozin downregulated the influenza virus-activated expression of IL-6, CCL2, and CCL5 in the RAW 264.7 cell line and in the macrophages isolated from the mouse bone marrow. The influenza virus upregulated the expression of TLR3, TLR7, and RIG-I involved in the recognition of the virus antigens and activation of the downstream signaling pathways in the cultivated RAW 264.7 macrophages, whereas empagliflozin decreased the level of TLR3 but not TLR7 and RIG-I [100]. In agreement, in the influenza virus-infected mice, empagliflozin decreased the TLR3 expression in the lungs but had no effect on the TLR7 and RIG-I expression. The authors assumed that the decrease in the macrophage proinflammatory response to the influenza virus caused by empagliflozin is associated with the inhibition of the TLR3 signaling pathway [100]. Considering our recent report that high glucose can upregulate TLRs on primary human macrophage ex vivo, it is of interest to examine whether SGLT2 inhibitors can interfere with such proinflammatory effects of hyperglycemia [66].

A human monocyte-like THP-1 cell line is another in vitro model that has been used for studying the anti-inflammatory effects of SGLT2 inhibitors. The experiments with this line provided rather contradictory results. On the one hand, canagliflozin demonstrated an inhibitory effect on the LPS-stimulated IL-1α production and secretion of TNF-α and IL-6 by THP-1 cells as well as by the RAW 264.7 cell line [111]. On the other hand, no suppressive effect of dapagliflozin, empagliflozin, and canagliflozin on the IL-6, IL-8, and IL-1β expression in the LPS-stimulated THP-1 cells was found [30]. It was hypothesized that the effects of SGLT2 inhibitors on macrophages in vivo may be mediated through a decrease in production of signaling molecules by other cells. However, THP-1 cells are very distant in their biology from primary monocyte-derived macrophages, since this cell line has a cancer origin, and THP-1 cells proliferate very efficiently, in contrast to primary human monocyte-derived macrophages. We have established a human primary monocyte-derived macrophage model to generate diabetic macrophages ex vivo [25,65,66], and such a model will enable us to identify the effects and mechanisms of action of SGLT2 inhibitors much more precisely.

In a model of primary mouse bone marrow cells stimulated with LPS, dapagliflozin caused the macrophage polarization towards the M2 phenotype (Arg-1+), while the number of M1 (iNOS+) cells decreased [90]. A conditioned medium of the macrophages cultivated with dapagliflozin activated the migratory and proliferative activities of the human umbilical vein endothelial cells (HUVECs) and enhanced formation of tubular structures. These data indicated that activation of the angiogenic potential in the presence of dapagliflozin can be mediated by macrophages [90].

Comparison of four SGLT2 inhibitors on the cultivated BV-2 mouse microglial cells in the high-glucose-induced inflammation model [113] showed that canagliflozin was the most potent in suppressing the secretion of proinflammatory molecules, such as COX-2, TNF-α, IL-1β, iNOS, and NLRP3, as well as in reduction of the formation of free oxygen radicals, providing the best protection for microglia against the high-glucose-induced inflammatory toxicity. An anti-inflammatory effect of empagliflozin was observed in the LPS-activated rat primary microglia model. The SGLT2 inhibitor decreased the expression of the Nos2, IL-6, TNF-α, and IL-1β [114]. In non-stimulated primary microglia cells, empagliflozin had no effect on the IL-6, TNF-α, and IL-1β expression.

In contrast to the majority of published reports, the proinflammatory effect of the SGLT2 inhibitor was observed in the mouse bone marrow-derived macrophages [115]. In this study, metformin and empagliflozin used as single agents activated the expression of Tnfa, Il1β, Il6, and Ifng genes in mouse macrophages; however, not all effects were increased with the increased dose of the agent. Moreover, the combination of metformin and empagliflozin abrogated some of the proinflammatory effects of a single agent. An in silico analysis of the protein–ligand interaction showed that empagliflozin interacted with both TLR2 and DECTIN1 receptors, while empagliflozin and metformin upregulated the expression of Tlr2 and Clec7a. The authors assumed that metformin and empagliflozin can directly modulate the expression of inflammatory genes in macrophages and boost the expression of receptors [115]. The discrepancy in the results of this study and other studies can be explained by the fact that Arefin and Gage analyzed the effect of gliflozins on nonpolarized macrophages, while other studies considered the effects on pre-stimulated macrophages. Another possibility is that the dose and time of inhibitor treatment can differ between various in vitro experimental setups.

In summary, the majority of in vitro studies in macrophage models correlate very well with the data obtained in a wide range of animal models, demonstrating the anti-inflammatory effects of SGLT2 inhibitors in diabetes and other diseases.

## 6. Molecular Mechanisms of the Anti-Inflammatory Effect of SGLT2 Inhibitors

### 6.1. Inflammasome Activity

In patients with type 2 diabetes, SGLT2 inhibitors decrease the levels of proinflammatory cytokines (TNF-α, IL-6, and IL-1β) and attenuate the activity of NLRP3 inflammasome [17,18,31]. Inflammasome is a key effector in the nonspecific immunity and is represented by an intracellular protein complex acting as a platform that manufactures proinflammatory cytokines, such as IL-1β [116]. Inflammasome comprises three components, namely, protein sensor (NLR and RLRs), protein adapters carrying a caspase recruitment domain (CARD), and procaspase-1. The best studied is inflammasome 3 (NLRP3), which belongs to the NOD-like receptor family and is a therapeutic target of SGLT2 inhibitors. An active inflammasome is assembled in response to the activation of signaling pathways by the danger signals PAMPs and DAMPs (pathogen- and damage-associated molecular patterns, respectively). The inactive pro-IL-18 and pro-IL-1β cytokines and an additional signal activate caspase, which leads to the formation of active proinflammatory cytokines and their secretion [116]. Analysis of macrophages from people with type 2 diabetes treated with empagliflozin or sulfonylurea demonstrated that the SGLT2 inhibitor was more efficient in the attenuation of NLRP3 activation in macrophages as compared to sulfonylurea, and more efficiently suppressed IL-1β secretion [31].

Dapagliflozin was shown to increase the amount of itaconate, an endogenous metabolite that inhibits the activation of NLRP3 inflammasome [32]. In macrophages, itaconate and its derivatives accumulate in response to M1 polarization. Itaconate activates proinflammatory transcription factors Nrf2 and ATF3, as well as inhibits the activation of NLRP3 inflammasome; its anti-inflammatory action was confirmed in preclinical models of sepsis, virus infections, psoriasis, gout, ischemia reperfusion injury, and pulmonary fibrosis [117].

### 6.2. Signaling Pathways in Macrophages Affected by SGLT2 Inhibitors

Although the molecular mechanisms underlying the anti-inflammatory effect of SGLT2 inhibitors in macrophages are still rather vague, the available data suggest that SGLT2 inhibitors can inhibit the MAPK, NF-κB, and JAK2/STAT1 signaling pathways. In particular, a study of the effect of empagliflozin on LPS-stimulated mouse RAW 264.7 macrophages promoting M1 macrophage polarization demonstrates that this inhibitor added simultaneously with LPS had a pronounced anti-inflammatory activity by decreasing production of prostaglandin E2 and proinflammatory cytokines. Empagliflozin inhibited phosphorylation of NF-κB, a crucial proinflammatory transcription factor in macrophages [118]. Empagliflozin was found to attenuate the phosphorylation of JNK without any effect on the p38 and ERK phosphorylation [112]. In the same cell model, dapagliflozin decreased NF-κB activation and expression of IL-6 and TNF-α [85]. A downregulation of toll-like receptor 4 (TLR4) gene expression was also recorded, suggesting that SGLT2 inhibitors can have a direct anti-inflammatory effect on macrophages by targeting the TLR4/NF-κB pathway [85].

It was demonstrated that canagliflozin inhibits IL-1α production in the LPS-stimulated human THP-1 and mouse RAW 264.7 cell lines [111]. It was assumed that the action of canagliflozin in these models is mediated by the inhibition of intracellular glycolysis, as well as promotion of autophagy, and IL-1 degradation mediated by the upregulated expression of p62. In its turn, the enhanced autophagy and elevated p62 level can be mediated by an increase in the activity of AMP-activated protein kinase (AMPK) and NF-κB, respectively [111]. AMPK is considered as a key regulator of the metabolic pathways controlling inflammation [119]. Empagliflozin was shown to restore the level of phosphorylated p-AMPK in mouse RAW 264.7 macrophages, which decreased in the presence of oxidized lipoproteins (ox-LDL) [120].

Dapagliflozin decreased the M1/M2 ratio and LPS-stimulated TLR4 expression in differentiated human monocyte-derived macrophages cultured in the presence of normal (5.5 mmol/L) or high glucose (25 mmol/L) [121]. Dapagliflozin also considerably lowered the phosphorylation of p65 NF-κB subunit in macrophages independently of the glucose level. Dapagliflozin abrogated the changes in the LPS-induced expression of miR-146a (anti-inflammatory) and miR-155 (proinflammatory). These miRs are important for NF-κB activation [121]. Dapagliflozin was shown to modulate the macrophage polarization, at least partially, by inhibiting the NF-κB signaling pathway. In a murine model of muscle ischemia, dapagliflozin reduced the NF-κB and I-κB phosphorylation [90].

An anti-inflammatory effect of empagliflozin was identified in an LPS-activated primary microglia rat model. The effect was associated with the inhibition of NF-κB and ERK1/2-MAPK signaling pathways and a decrease in the expression of proinflammatory mediators Nos2, IL-6, and TNF-α [114]. A low basal activity of the NF-κB signaling pathway in microglial cells was demonstrated, and this activity was considerably increased during the immune response [122]. Members of the MAP kinase family are activated in microglial cells by different inflammatory signals and are involved in cell-mediated immunity and self-degradation [123]. Effects of four SGLT2 inhibitors (canagliflozin, dapagliflozin, empagliflozin, and etrugliflozin) were assessed in the cultivated BV-2 mouse microglial cells in a high-glucose-induced inflammation model [113]. Among the four drugs analyzed, canagliflozin inhibited high-glucose-induced secretion of the proinflammatory molecules (COX-2, TNF-α, IL-1β, iNOS, and NLRP3) more markedly. The mechanism of the canagliflozin effect involved a decrease in the activity of high-glucose activation of NF-κB, JNK, p38, and PI3K/Akt signaling pathways, where the PI3K/Akt pathway controls both inflammation and cell survival [113,124].

In macrophages, the STAT family of transcription factors controls production of different cytokines and growth factors with both pro- and anti-inflammatory activities [125]. STATs can be activated via phosphorylation resulting from the activation of Janus kinases (JAKs) by proinflammatory signals, in particular, LPS. Empagliflozin was shown to inhibit the LPS-induced JAK2 phosphorylation in RAW 264.7 macrophages as well as STAT1 phosphorylation [112]. The potential of dapagliflozin to regulate the macrophage polarization by the activation of STAT3 was observed in infarcted rat hearts [33]. In this model, M1 macrophages accumulated in the myocardium on day 3 after infarction. Under dapagliflozin treatment, an increase in the M2/M1 ratio towards the M2c subtype (CD68+ and IL-10+) was found. The mRNA levels of proinflammatory M1-like cytokines (IL-6, IL-1β, and iNOS) were considerably decreased, while the expression levels of CD206 and IL-10, characteristic for M2, were elevated. Activation of STAT3 was critical for the anti-inflammatory effects of dapagliflozin, since an inhibitor of STAT3 signaling (S3I-201) abrogated macrophage polarization to M2c in the presence of dapagliflozin in infarcted rat hearts [33]. In a mouse model of acute viral myocarditis, dapagliflozin activated STAT3 phosphorylation, increased an anti-inflammatory macrophage polarization, and mitigated inflammation. A simultaneous administration of dapagliflozin and STATTIC, a STAT3 inhibitor, canceled the anti-inflammatory effect of dapagliflozin [34].

The molecular mechanisms of the anti-inflammatory effect of SGLT2 inhibitors on macrophages are summarized in Figure 3. A growing body of evidence suggests that the effect of these therapeutic agents involves an attenuation of inflammasome activity and inhibition of TLR4/NF-κB. The anti-inflammatory effect is also related to the changes in the activity of other signaling pathways (AMPK, PI3K/Akt, ERK 1/2-MAPK, and JAKs/STAT) in macrophages.

### 6.3. SGLT2 Control Metabolic Inflammation in Multiple Cell Types

SGLT2 inhibitors exert diverse effects on human health, not only by direct mechanistic changes in macrophage signaling and metabolism, but also by affecting the biology of other cells types that interact with macrophages in pathologies related to metabolic inflammation [126]. These effects of SGLT2 inhibitors are shown in Figure 4.

Notably, the action of SGLT2 inhibitors extends far beyond their primary role in inhibiting glucose reabsorption in the kidneys [126,127,128,129]. SGLT2 inhibitors influence vascular health through multiple mechanisms, controlling deceleration of age-related vascular changes [127,128]. These agents have demonstrated significant anti-aging effects on vascular smooth muscle cells (VSMCs) in diabetic animal models [127,128]. SGLT2 inhibitors are able to restore damaged VSMCs to a healthy state, normalize smooth muscle cell function, and prevent endothelial dysfunction [127]. These beneficial effects are attributed to the inhibitor’s ability to reduce glucose toxicity, oxidative stress, and inflammation while promoting endothelial cell viability [126,127,128]. Dapagliflozin effectively prevented oxidative stress in lens epithelial cells of diabetic mice [130]. By blocking the activation of NADPH oxidase, a key enzyme involved in ROS production, dapagliflozin reduced oxidative damage and protected these cells [130]. This effect was likely mediated by its ability to decrease glucose uptake and ROS generation [130]. Furthermore, SGLT2 inhibitors delay vascular aging, restore vascular smooth muscle cell function, and reduce chronic inflammation by promoting anti-inflammatory macrophage polarization and blocking inflammatory pathways [131].

In atherosclerosis, a major macrovascular complication of diabetes, SGLT2 inhibitors have a significant impact. For example, empagliflozin and canagliflozin can delay disease progression by reducing hyperglycemia, hyperlipidemia, and inflammation, improving hemodynamic parameters, and stabilizing atherosclerotic plaques [132,133]. A study that involved patients with type 2 diabetes revealed that dapagliflozin significantly reduced carotid–femoral artery pulsation, indicating its potential to alleviate long-term arterial stiffness [134]. Moreover, SGLT2 inhibitors enhance the production of nitric oxide, a key molecule that relaxes blood vessels, restoring normal blood vessel function in people with diabetes [135]. This effect reduced carotid–femoral pulse wave velocity, independent of blood pressure changes [134]. Another study compared empagliflozin, metformin, and their combination in type 1 diabetes patients, showing that empagliflozin was more effective than metformin in reducing arterial stiffness, with combined use yielding even better results [136]. Interestingly, empagliflozin’s effect appears to be independent of the endothelial function, possibly involving specific receptor signaling pathways [136]. Dapagliflozin also improved arterial vasodilation and systemic endothelial function in patients with type 2 diabetes, while canagliflozin inhibited vascular smooth cell proliferation [137].

High glucose levels have been shown to induce epithelial–mesenchymal transition (EMT) in renal tubular epithelial cells [138]. This process facilitates the progression of renal fibrosis, particularly in diabetic conditions, as EMT contributes to the transformation of epithelial cells into mesenchymal cells, leading to the tissue scarring and loss of function [139,140]. The critical role of EMT in accelerating renal fibrosis justifies the importance of therapeutic interventions in mitigating this pathway [139,140,141,142]. Empagliflozin and canagliflozin are promising therapeutics since they effectively suppress glucose-induced EMT and reduce renal fibrosis in diabetic mice [138]. Furthermore, the significance of this suppressive mechanism has even more impact if we consider the role of Angiotensin II (AngII) in inducing EMT and pro-fibrotic pathways in human kidney cells (HK-2) [143]. By decreasing the expression of critical proteins such as SIRT3, FOXO3a, and catalase, AngII exacerbates renal damage [143]. Canagliflozin is able to reverse these detrimental effects, protecting kidneys from fibrogenesis [143]. The silencing of SIRT3 blocked the ability of canagliflozin to restore FOXO3a and catalase levels, preventing its protective effect against EMT [143]. These findings argue towards the mechanism of canagliflozin action via the activation of the SIRT3-FOXO3a pathway, thereby suppressing EMT and potentially preventing fibrosis, as demonstrated in high-salt diet-induced Dahl salt-sensitive rats [143]. In addition to these effects, dapagliflozin has been found to prevent endothelial–mesenchymal transition in tubular epithelial cells within the peritubular capillaries [144]. By targeting these critical interactions, SGLT2 inhibitors offer a broader protective mechanism against renal damage and fibrosis. Their ability to suppress EMT, to restore protective protein levels, and at the same time to suppress the resident macrophage-mediated local inflammation reinforces the role of SGLT2 inhibitors the prevention of kidney disease progression, especially under diabetic conditions.

SGLT2 inhibitors have been shown to influence the calcification of smooth muscle cells, a process that can be facilitated by altered mineral metabolism, oxidative stress, and inflammation [145,146,147]. These factors play a critical role in the development of vascular calcification, which can contribute to vascular stiffness and related complications. In diabetes, vascular calcification can further exacerbate the risk of cardiovascular complications, while inflammatory macrophages can elevate the calcification level [147,148]. Empagliflozin was shown to reduce atherosclerotic calcification in ApoE^−/−^ mice [145]. This effect can include both direct targeting of vascular cells as well as indirect reduction of macrophage-mediated inflammation. By inhibiting the osteogenic differentiation of vascular smooth muscle cells (VSMCs), empagliflozin significantly reduced calcification, both in vivo and in vitro, as evidenced by its effects in VSMCs and aortic rings [145]. These data highlight the potential of empagliflozin as a therapeutic option for managing vascular calcification [145]. Similarly, canagliflozin has been shown to prevent aortic calcification by targeting the NLRP3 inflammasome pathway [146]. This pathway plays a key role in mediating inflammation in different cells types, including macrophages, and its inhibition by canagliflozin resulted in reduced calcification in isolated VSMCs [146]. Additionally, canagliflozin significantly decreased the expression of NLRP3 and its downstream signaling molecules, such as caspase-1 and IL-1β [146]. Independent studies further confirmed the importance of the NLRP3 pathway [146]. VSMC calcification was reduced when NLRP3 inhibitor (MCC950) and siRNA knockdown of NLRP3 were applied [146]. Conversely, activating NLRP3 exacerbated calcification in VSMCs, but this effect was reversed by canagliflozin treatment [146]. Moreover, in a study involving 1554 patients with type 2 diabetes, computed tomography angiography revealed that the use of SGLT2 inhibitors, such as dapagliflozin, was associated with a lower Agatston calcification score, a measure of coronary artery calcification [149]. These clinical findings are supported by the experiments in animal models, where dapagliflozin significantly reduced vascular calcification in the aorta [149]. The protective effect of dapagliflozin against the calcification was mediated through the downregulation of thioredoxin domain containing 5 (TXNDC5) in VSMCs [149]. This downregulation disrupted the stability of Runx2, a key transcription factor involved in osteogenic differentiation, leading to its proteasomal degradation [149]. The ability of dapagliflozin to downregulate TXNDC5 is believed to result from its effects on reducing oxidative stress and endoplasmic reticulum stress [149].

SGLT2 inhibitors have been shown to impact kidney proximal tubule cell migration, potentially by modulating various cellular pathways involved in cell motility [150]. These effects may be linked to alterations in glucose transport, oxidative stress reduction, and inflammatory signaling, which could influence the repair processes in kidney tissues [150,151]. By affecting these mechanisms, SGLT2 inhibitors play a role in improving kidney function and mitigating damage in conditions such as diabetic nephropathy [150]. Empagliflozin exerted the renoprotective effects in high-glucose-induced cultured kidney proximal tubule cells in vitro [152]. Empagliflozin-attenuated high-glucose-induced oxidative stress, MMP2 activation, and repression of reversion-inducing cysteine-rich protein with kazal motifs (RECK) which were associated with suppressed EMT and migration of kidney proximal tubule cells [152]. These effects are linked to reductions in the expression of several proinflammatory and pro-fibrotic mediators, including TRAF3IP2, NF-κB, p38MAPK, miR-21, IL-1β, IL-6, TNFα, and MCP1 in kidney proximal tubule cells [152]. Empagliflozin prevented advanced glycation end product (AGE)/receptor for AGE (RAGE)-induced TRAF3IP2 expression and RECK suppression in kidney proximal tubule cells [152]. In an in vitro scratch assay using HK2 cells, canagliflozin was found to have a similar effect to empagliflozin in suppressing kidney proximal tubule cells migration. HK2 cells were cultured under high-glucose conditions (25 mmol/L) for 24 h, which promoted the migration of cells into the scratch-free area [153]. Treatment with canagliflozin significantly suppressed this migration, indicating that both SGLT2 inhibitors have the ability to inhibit the migration kidney proximal tubule cells induced by high glucose [153]. High-mobility group box 1 (HMGB1) is a protein that is involved in various cellular processes, including inflammation, cell proliferation, and differentiation [154,155,156]. In the context of diabetic nephropathy, HMGB1 has been shown to play a role in the development and progression of renal injury [157]. The SGLT2 inhibitor dapagliflozin can reduce HMGB1 levels and improve renal function in diabetic nephropathy [158]. Dapagliflozin treatment reduced expression and secretion of HMGB1 in cultured kidney proximal tubule cells [158]. Dapagliflozin also reduced the levels of inflammatory markers, such as monocyte chemoattractant protein-1 (MCP-1) and intercellular adhesion molecule-1 (ICAM-1), and oxidative stress indicators, such as malondialdehyde (MDA) and superoxide dismutase (SOD) in human renal proximal tubular cells [158]. Furthermore, dapagliflozin treatment inhibited the activation of the HMGB1 receptor for advanced glycation end products (RAGE)-nuclear factor-kappa B (NF-κB) signaling pathway, which is involved in inflammation and fibrosis [158]. Dapagliflozin may delay tubulointerstitial fibrosis in diabetic kidney disease, inhibiting YAP/TAZ activation in tubular epithelial cells [159]. In a mouse model of diabetic kidney disease, the SGLT2 inhibitor dapagliflozin was found to reduce the activation of YAP/TAZ pathway in human proximal tubular epithelial cells and to suppress expression of their target genes, connective tissue growth factor (CTGF) and amphiregulin, both involved in fibrosis [159]. Importantly, dapagliflozin inhibited inflammation, oxidative stress, and fibrosis in the kidneys in the rat model for diabetic kidney disease [159].

Glucose is a vital nutrient in the bloodstream, filtered in the kidneys across the glomeruli and reabsorbed back into the blood. This reabsorption is facilitated by the transporters SGLT2 and SGLT1, which are found in the apical membrane of the renal proximal convoluted tubules [160,161]. Specifically, these transporters utilize the sodium ion electrochemical gradient to enable glucose uptake. SGLT2 is predominantly located in the S1 and S2 segments of the proximal tubules and is responsible for 90% of glucose reabsorption, whereas SGLT1 accounts for the remaining 10% [161]. Furthermore, SGLT2 cotransporters are also present in various other tissues, including the intestine, kidneys, glands, testis, liver, lung, skeletal muscle, spleen, and cerebellum, highlighting their crucial role in facilitating glucose absorption beyond the kidneys [162] (Table 3).

In summary, SGLT2 inhibitors exert a wide range of effects on key cell types responsible for diabetic macro- and microvascular complications. These effects include improving vascular health, reducing vascular calcification, and preventing endothelial dysfunction. In diabetic kidney disease models, SGLT2 inhibitors like dapagliflozin and empagliflozin have shown significant anti-inflammatory, anti-oxidative, and anti-fibrotic properties. They reduce calcification in vascular smooth muscle cells, inhibit epithelial–mesenchymal transition, and delay fibrosis by modulating pathways such as YAP/TAZ, TXNDC5, and NLRP3. These inhibitors also improve endothelial function and arterial stiffness, with benefits observed in both human and animal studies. Furthermore, SGLT2 inhibitors impact kidney function by regulating cell migration, reducing oxidative stress, and mitigating the effects of high glucose levels on proximal tubule cells, showing potential as therapeutic agents for diabetic nephropathy. The reduction of macrophage-mediated inflammation in the target organs has a synergistic effect and amplifies the therapeutic power of SGLT2 inhibitors.

## 7. Conclusions

Accumulating evidence from clinical and experimental studies indicates the potential of SGLT2 inhibitors to reduce low-grade inflammation in diabetes and other human diseases. Suppression of the proinflammatory programming of tissue-resident macrophages as well as recruited monocyte-derived macrophages by SGLT2 inhibitors plays an important role in the anti-inflammatory effect of these drugs. The molecular mechanisms underlying the effects of gliflozins on macrophages are emerging; however, precise molecular interactions remain to be identified. Growing number of reports show suppression of inflammasome activity, inhibition of TLR4/NF-κB and other signaling pathways, and a decrease in production of proinflammatory cytokines. The finding that SGLT2 inhibitors instruct macrophages to shift to anti-inflammatory oxidative phosphorylation metabolism is intriguing and raises questions about the molecular links between signaling and metabolic pathways. Multiple studies indicate that SGLT2 inhibitors improve endothelial and epithelial cell functions in the context of the macrophage-mediated inflammation. Identification of the crosstalk between cells responding to SGLT2 inhibitors in the target organs is a challenging task.

Substantial progress in understanding the mechanisms of SGLT2 inhibitors’ action can be expected in the coming years. Rapid development of the high-throughput gene expression analysis, such as RNA-seq, ChIP-seq, and single-cell transcriptome sequencing, can provide fundamentally new data on the modifying effect of SGLT2 inhibitors on various subpopulations of macrophages involved in the processes of inflammation, fibrosis, and tissue repair. Omics data may contribute to the identification of genetic and epigenetic mechanisms of the systemic effects of SGLT2 inhibitors, including effects on innate and adaptive immune cells. Identification of key molecules and signaling pathways that define the mechanisms of protective effects of SGLT2 inhibitors on target organs may contribute to the development of new approaches to the treatment of diabetes and other common diseases.

## Figures and Tables

**Figure 1 ijms-26-01670-f001:**
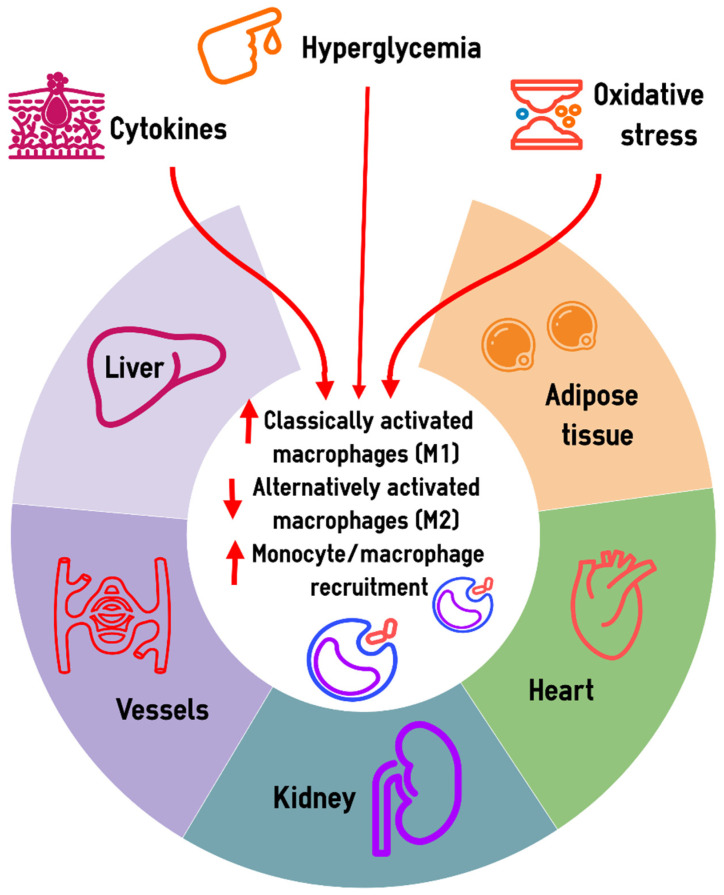
Macrophages in diabetes-related chronic low-grade inflammation. Hyperglycemia, increased production of proinflammatory cytokines, and oxidative stress are the main drivers of macrophage changes in diabetes. These factors determine the predominance of classically activated macrophages (M1) over alternatively activated macrophages (M2), and also enhance the recruitment of monocytes/macrophages in adipose tissue, myocardium, kidneys, blood vessels, and the liver. Macrophage activation is a hallmark of chronic low-grade inflammation, which is involved in the pathogenesis of diabetic complications.

**Figure 2 ijms-26-01670-f002:**
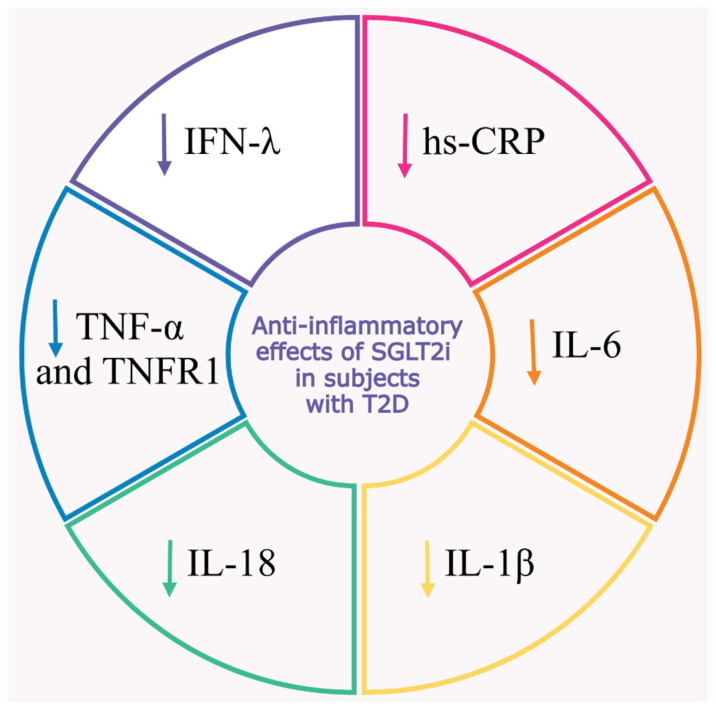
Effects of sodium–glucose contransporter 2 inhibitors (SGLT2i) on circulating inflammatory markers in patients with type 2 diabetes (T2D). Treatment with SGLT2i is associated with decreased serum levels of high-sensitivity C-reactive protein (hs-CRP), a marker of low-grade inflammation, in patients with T2D. The use of SGLT2i is also associated with decreased concentrations of circulating proinflammatory cytokines, such as interleukin-6 (IL-6), interleukin-1 β (IL-1β), interleukin-18 (IL-18), tumor necrosis factor α (TNF-α), tumor necrosis factor receptor 1 (TNFR1), and interferon- λ (INF-λ). These molecules are widely discussed as drivers and biomarkers of diabetic vascular complications.

**Figure 3 ijms-26-01670-f003:**
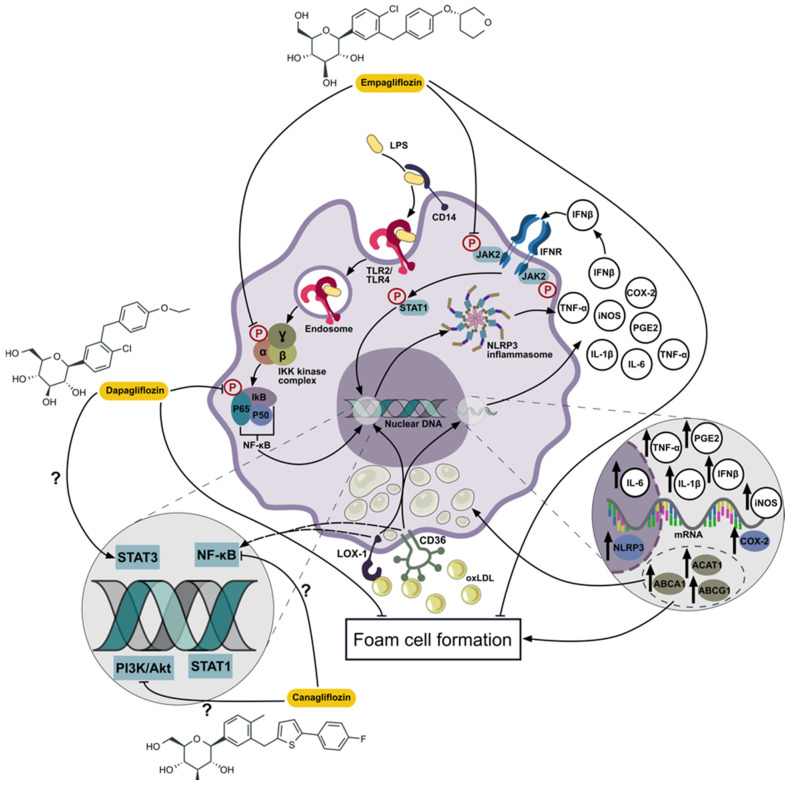
Molecular mechanisms of SGLT2 inhibitors in modulating anti-inflammatory signaling pathways in macrophages. Overview of SGLT2 inhibitors, including empagliflozin, dapagliflozin, and canagliflozin, on key macrophage anti-inflammatory signaling pathways. SGLT2 inhibitors interact with various components of the inflammatory signaling cascade, including TLR2/TLR4, JAK2/STAT1, and the NLRP3 inflammasome, suppressing activation of NF-κB and PI3K/Akt signaling pathways while activating STAT3. These interactions lead to alterations in the expression of proinflammatory cytokines (e.g., TNF-α, IL-1β, IL-6, IFN-β), inducible nitric oxide synthase (iNOS), and cyclooxygenase-2 (COX-2), which contribute to the anti-inflammatory response and suppression of foam cell formation.

**Figure 4 ijms-26-01670-f004:**
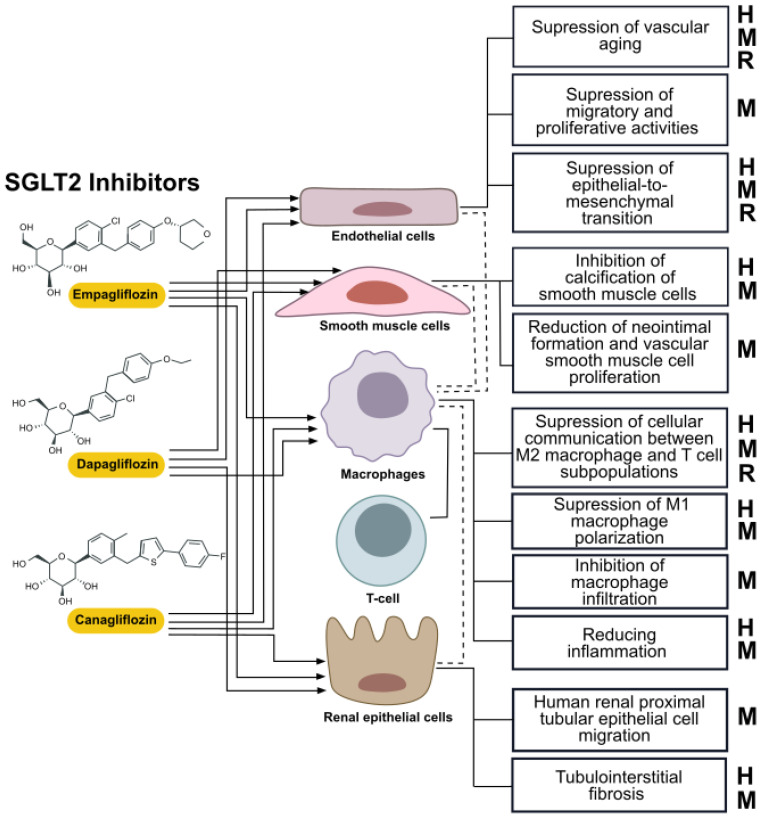
SGLT2 inhibitors reduce inflammation by affecting crosstalk between macrophage and other cell types. The figure illustrates the cellular targets and mechanisms influenced by SGLT2 inhibitors across different cell types, including endothelial cells, smooth muscle cells, macrophages, T-cells, and renal epithelial cells. This summary emphasizes the potential of SGLT2 inhibitors to influence not only glucose metabolism but also vascular and immune cell functions, contributing to their therapeutic benefits in cardiovascular and renal protection. The letters (H, M, and R) indicate which the findings are based on human (H), mouse (M), or rat (R) samples or models.

**Table 1 ijms-26-01670-t001:** Anti-inflammatory effects of SGLT2 inhibitors in subjects with type 2 diabetes.

Study	Inhibitor	Patients	Anti-Inflammatory Effect	Ref.
EMPA-REG OUTCOME	Empagliflozin	T2D, high cardiovascular risk	Decrease in hs-CRP concentrations after 12 months of treatment.	[72]
CANOSSA	Canagliflozin	T2D, chronic heart failure	Decrease in hs-CRP concentrations after 3, 6, and 12 months of treatment.	[21,73]
Iannantuoni et al. (2019)	Empagliflozin	T2D	Decrease in hs-CRP and increase in IL-10 concentrations after 24 weeks of treatment.	[74]
La Grotta et al. (2022)	Dapagliflozin, empagliflozin, canagliflozin	T2D	Lower plasma IL-6 levels in patients treated with SGLT2 inhibitors compared with those treated with other antihyperglycemic agents.	[30]
Gotzmann et al. (2023)	Empagliflozin	T2D, chronic heart failure	Decrease in serum IL-6 concentrations after 3 month follow-up in empagliflozin-treated cohort.	[75]
CANTATA-SU	Canagliflozin	T2D	Decrease in plasma TNFR1 and IL-6 levels during 104 weeks of treatment in canagliflozin cohort when compared to glimepiride cohort.	[76]
Kim et al. (2020)	Empagliflozin	T2D, high cardiovascular risk	Reduced production of IL-1β and TNF-α by macrophages in empagliflozin-treated subjects.	[31]
CANVAS	Canagliflozin	T2D	Decrease in plasma TNFR1 and TNFR2 concentrations after years 1, 3 and 6 of treatment.	[77]
Meta-analysis of 18 randomized clinical trials	Dapagliflozin, empagliflozin, canagliflozin	T2D	Treatment with SGLT2 inhibitors is associated with lower serum IL-6 levels.	[29]

CANOSSA, prospective, open-label, add-on trial of canagliflozin for diabetes mellitus and stable chronic heart failure; CANTATA-SU, CANagliflozin Treatment And Trial Analysis-Sulfonylurea SGLT2 Add-on to Metformin Versus Glimepiride; CANVAS, CANagliflozin cardioVascular Assessment Study; EMPA-REG OUTCOME, BI 10,773 (Empagliflozin) Cardiovascular Outcome Event Trial in Type 2 Diabetes Mellitus Patients; hs-CRP, high-sensitivity C-reactive protein; IL-1β, interleukin-1 beta; IL-6, interleukin 6; IL-10, interleukin 10; T2D, type 2 diabetes; TNF-α, tumor necrosis factor alpha; TNFR1, tumor necrosis factor receptor 1; TNFR2, tumor necrosis factor receptor 2.

**Table 2 ijms-26-01670-t002:** Anti-inflammatory effects of SGLT2 inhibitors in the animal models.

SGLT2 Inhibitor	Animals	Disease Model	Anti-Inflammatory Effects of SGLT2 Inhibitors	Ref.
*Animal models for obesity and diabetes*				
1 Empa	C57BL/6J mice	Streptozotocin-induced diabetes Atherosclerosis induced by LDLR- and SRB1- antisense oligonucleotides and high cholesterol diet	Decreased content of CD68+ macrophages in atherosclerotic plaques Decreased proliferation of plaque resident macrophages in atherosclerotic plaques	[84]
2 Ipra	C57BL/6 J CCR2 knockout mice	High-fat diet-induced obesity	Reduction of M1/M2 ratio in adipose tissue	[95]
3 Empa	C57BL/6J mice	High-fat diet-induced obesity	M2 polarization in white adipose tissue and liver	[97]
4 Empa	APP/PS1× *db/db* mice	Alzheimer’s disease Genetically determined obesity and Type 2 diabetes	Mitigation of microglia damage in senile plaque-free areas	[92]
5 Empa, Dapa	CD-1 mice	Streptozotocin-induced diabetes	Decreased IL-1β and IL-6 levels in prefrontal cortex	[38]
6 Dapa	LDL-receptor-deficient (Ldlr^−/−^) mice	High-fat high-sucrose diet-induced Type 2 diabetes	Reduction of circulating platelet-leukocyte aggregates; decreased aortic macrophage infiltration	[107]
7 Dapa Ipra	Apoe^−/−^ mice *db/db* mice	Streptozotocin-induced diabetes Genetically determined obesity and Type 2 diabetes	Attenuated macrophage infiltration in the aortic root Upregulation of lectin-like ox-LDL receptor-1 and acyl-coenzyme A:cholesterol acyltransferase 1 gene expression in the peritoneal macrophages Downregulation of ATP-binding cassette transporter A1 in the peritoneal macrophages	[86]
8 Empa	Zucker diabetic fatty rats	Obesity and Type 2 diabetes	Reduction of the expression of IL-1β, IL-6, TNF-α, MCP-1 and IL-10 in visceral adipose tissue Decreased expression of M1 and M2 markers in visceral adipose tissue	[108]
9 Dapa	New Zealand white rabbits	Alloxan-induced diabetes	Decreased macrophages infiltration and expression of TNF-α in the myocardium	[109]
10 Empa	C57BL/6J mice	High-fat diet and streptozotocin-induced diabetes	Activation of autophagy in hepatic macrophages via the AMPK/mTOR signaling pathway Decreased expression levels of IL-17/IL-23 axis-related molecules in the liver	[110]
*Non-diabetic animal models*				
1 Dapa	New Zealand white rabbits	Atherosclerosis	Mitigation of macrophage infiltration in atherosclerotic plaques Decreased expression of TNF-α, IL-1β and IL-6 in atherosclerotic plaques Suppression of TLR4/NF-κΒ pathway in RAW 264.7 macrophages Increased M2 levels in atherosclerotic plaques	[85]
2 Empa	C57BL/6 mice LDLR−/− mice	Influenza A/PR/8/34 High-fat diet-induced atherosclerosis	Decreased expression of IL-1β, IL-6, and CCL2 in the lungs Decreased expression of IL-1β, IL-6, and CCL2 in RAW264.7 macrophages and bone marrow-derived macrophages Decreased percentage of inflammatory monocytes and inducible NO synthase-positive macrophages in the lungs Decreased expression of Stat1 and CXCL9 in macrophages in response of IFN exposure	[100]
3 Dapa	C57BL/6 mice	Critical limb ischemia	M2 polarization in gastrocnemius muscle Diminished expression IL-1β, IL-6, and TNF-α in gastrocnemius muscle	[90]
4 Dapa	Balb/c mice	Coxsackie virus B3-induced acute viral myocarditis	Decreased IL-1β, IL-6, and TNF-α levels in the myocardium Inhibition of M1 and promotion of M2 differentiation in the myocardium Activation of Stat3 pathway	[34]
5 Empa	Spontaneously hypertension rats	Arterial hypertension	Decreased renal expression of MAPK10 Decreased pulmonary expression Legumain and cathepsin S (CTSS) Increased expression of Prkaa1 and Prkaa2 in white adipose tissue	[36]
6 Dapa	C57BL/6J mice	Ischemia/reperfusion induced renal fibrosis	Activation of mTOR and HIF-1α pathways in the renal cortex Blocked activation of NLRP3 inflammasome in the renal cortex	[32]
7 Cana	C57BL/6 mice	LPS-induced acute lung injury	Decreased levels of TNF-α, IL-6, and IL-1β in bronchoalveolar lavage fluid and serum Reduction of alveolar macrophages with the M1 phenotype Promotion of shift of alveolar macrophages towards M2	[99]
8 Empa	Wistar rats	Acute acetic acid-induced ulcerative colitis	Decreased serum C-reactive protein levels Induced SIRT-1 expression in colonic tissues Reduced PI3K, AKT, NF-κB, TNF-α, IL-1β, and IL-6 expression in colonic tissues	[104]
9 Empa	Wistar rats	Dextran sulfate sodium-induced ulcerative colitis	Enhanced AMPK phosphorylation in distal colons Depressed mTOR and NLRP3 expression in distal colons Reduction in caspase-1 cleavage in distal colons Decreased expression of IL-1β and IL-18 in distal colons Reduction in Th17 cell polarization and maintenance in distal colons	[105]

Cana, canagliflozin; Dapa, dapagliflozin; Empa, empagliflozin; Ipra, ipragliflozin.

**Table 3 ijms-26-01670-t003:** Role of SGLT2 in different cell types and tissues.

Cell/Tissue Type	Role of SGLT2	Effects of SGLT2 Inhibition
Proximal Renal Tubule Cells	glucose reabsorption	– Reduction of glucose reabsorption – Attenuation of high-glucose-induced oxidative stress – Suppression of EMT and cell migration – Reduction expression of proinflammatory and pro-fibrotic mediators – Increasing of AMPK activity and maintaining cell viability
Vascular Smooth Muscle Cells	Glucose uptake and metabolism	– Restoring damaged VSMCs to a healthy state – Normalization of smooth muscle cell function – Inhibition of osteogenic differentiation and calcification – Reduction of atherosclerotic calcification
Endothelial Cells	Glucose transport and vascular function	– Limitation of endothelial dysfunction – Enhancing of nitric oxide production – Improving of arterial vasodilation and systemic endothelial function
Macrophages	Involvement in inflammation	– Promotion of anti-inflammatory macrophage polarization – Blocking of inflammatory pathways – Foam cell formation
Lens Epithelial Cells	Glucose uptake	– Improving of oxidative stress by blocking NADPH oxidase activation
Tubular Epithelial Cells	N/A	– Improving of endothelial–mesenchymal transition in peritubular capillaries – Inhibition of YAP/TAZ activation, suppressing fibrosis-related gene expression
Adipose Tissue	N/A	– Reduction of senescent cell accumulation and improve metabolic dysfunction
Cardiomyocytes	N/A	– Potential cardioprotective effects

## Data Availability

Not applicable.

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
