# Peer review of "Anti-Inflammatory Effects of SGLT2 Inhibitors: Focus on Macrophages"

_ijms, 2025, doi:10.3390/ijms26041670_

Round 1

Reviewer 1 Report

Comments and Suggestions for Authors

This review is comprehensive, interesting and overall well written. The effect of the SGLT2 inhibitors is relatively new and already quite accepted in heart failure, diabetes and some cancers as conforted by several clinical trials. 

You summarize some of the clinical studies related to the effect of the inhibitors on T2D (lines 177-180). It would useful for the readers to have a more comprehensive review of the SGLT2 inhibitors in clinical trials gathered in a Table if more such trial results are available. 

This reviewer has a general question of the hypothesized/ proposed mechanism of action, based on a reduction of inflammation / inflammatory cytokines etc. 

1) If this is the main mechanism of action, why the SGLT2 inhibitors have no action on hospitalized patients for severe COVID  (see for example Lancet Diabetes Endocrinol 2024 Oct Vale et al; Kosiborod et al; Sheen) while metformin, an antibiabetic drug was found protective in severe COVID (Ann Med 2024 Somasundara et al) ?? could you please comment 

Could it be related to the lung expression of the SGLT2 vs other organs as heart/ kidney ? 

2)  the author discuss the comparison between SGLT2 inhibitors and metformin in lines 369-386. Could you please extend the discussion in a broader sense? 

 Overall, I recommend minor revisions. 

Author Response

Reviewer 1

Comment: This review is comprehensive, interesting and overall well written. The effect of the SGLT2 inhibitors is relatively new and already quite accepted in heart failure, diabetes and some cancers as conforted by several clinical trials. 

You summarize some of the clinical studies related to the effect of the inhibitors on T2D (lines 177-180). It would useful for the readers to have a more comprehensive review of the SGLT2 inhibitors in clinical trials gathered in a Table if more such trial results are available. 

Answer: Thank you for this suggestion. To date, a large number of clinical studies of SGLT2

inhibitors have been conducted in both diabetic and non-diabetic populations. It is difficult to

summarize the results of these numerous studies in a single table. To maintain the focus of the

review and to improve readability, we have included in the revised version of the manuscript a

table summarizing the results of studies that have examined the anti-inflammatory effects of

SGLT2 inhibitors in patients with type 2 diabetes. This information is summarized in new

Table1 and discussed in the corresponding text fragment.

Comment:  If this is the main mechanism of action, why the SGLT2 inhibitors have no action on hospitalized patients for severe COVID  (see for example Lancet Diabetes Endocrinol 2024 Oct Vale et al; Kosiborod et al; Sheen) while metformin, an antidiabetic drug was found protective in severe COVID (Ann Med 2024 Somasundara et al) ?? could you please comment 

Answer: We completely agree with the reviewer's comment. Indeed, SGLT2 inhibitors have

Not shown clinical effects in COVID-19, unlike metformin. The mechanisms underlying these

differences are unknown, but confounder effects cannot be ruled out. We propose that the

mechanisms of anti-inflammatory action of SGLT2 inhibitors are secondary to their metabolic

effects. SGLT2 inhibitors may have an effect on metabolically-associated low-grade

inflammation. However, in high-grade inflammation (e.g., COVID-19), the anti-inflammatory

activity of these drugs is insufficient to provide a clinically significant result. We have included

this hypothesis in Chapter 3.

Comment: Could it be related to the lung expression of the SGLT2 vs other organs as heart/ kidney

Answer: we are grateful for the reviewer for this suggestion, and we believe that this idea needs experimental confirmation in future

Comment:  the author discuss the comparison between SGLT2 inhibitors and metformin in lines 369-386. Could you please extend the discussion in a broader sense? 

Answer: we are grateful to the review for raising this question, and we believe that comparison of the effects of SGLT2 and metformin needs deep experimentation, where mechanism of action and biological effects of these inhibitors will be compared in parallel in the same model systems by deep analytical methods like metabolomics identified by mass spectrometry and transcriptomics.

 Comment: Overall, I recommend minor revisions. 

Answer: we are grateful the reviewer for positive evaluation and highly valuable comments and ideas

Reviewer 2 Report

Comments and Suggestions for Authors

The manuscript describes the role of SGLT2 inhibitors in the decrease of inflammation promoted by macrophages dysfunction. The manuscript is easy to follow, and it is within the scope of the IJMS. However, Authors it is necessary to clearly delimit the article to the non-glycemic effects of SGLTi on inflammation and macrophages. What is the new in this manuscript? In addition, Authors supported their manuscript in an important quantity of reviews, instead of the original articles.

I suggest discussing the mechanisms involved in the intracellular signaling induced by SGLT2i. Are there SGLT2 transporters in macrophages, muscle cells, endothelial cells, among others? Is the transport coupled with scaffold proteins responsible of the intracellular signaling? What is the opinion of the expert?

The manuscript has typing mistakes.

Author Response

Reviewer 2.

Comment: The manuscript describes the role of SGLT2 inhibitors in the decrease of inflammation promoted by macrophages dysfunction. The manuscript is easy to follow, and it is within the scope of the IJMS. However, Authors it is necessary to clearly delimit the article to the non-glycemic effects of SGLTi on inflammation and macrophages.  

Answer: we thank the reviewed for the positive evaluation. We are convince that by presentation of the non-glycaemic effects of SGLT2 it is unavoidable to mention other effects and compare the mechanisms in order to place non- glycemic effects in the context of the complexity of SGLT2 inhibitor action on the immune system.

Comment: In addition, Authors supported their manuscript in an important quantity of reviews, instead of the original articles.

Answer: We have substantially improved the reference list and added essential original articles, as requested. Following original articles were added:

46.Stein, M.; Keshav, S.; Harris, N.; Gordon, S. Interleukin 4 Potently Enhances Murine Macrophage Mannose Receptor Activity: A Marker of Alternative Immunologic Macrophage Activation. J Exp Med 1992, 176, 287–292, doi:10.1084/jem.176.1.287.

  1. Ruffell, D.; Mourkioti, F.; Gambardella, A.; Kirstetter, P.; Lopez, R.G.; Rosenthal, N.; Nerlov, C. A CREB-C/EBPbeta Cascade Induces M2 Macrophage-Specific Gene Expression and Promotes Muscle Injury Repair. Proc Natl Acad Sci U S A 2009, 106, 17475–17480, doi:10.1073/pnas.0908641106.
  2. Vogel, D.Y.S.; Glim, J.E.; Stavenuiter, A.W.D.; Breur, M.; Heijnen, P.; Amor, S.; Dijkstra, C.D.; Beelen, R.H.J. Human Macrophage Polarization in Vitro: Maturation and Activation Methods Compared. Immunobiology 2014, 219, 695–703, doi:10.1016/j.imbio.2014.05.002.
  3. Parrinello, C.M.; Lutsey, P.L.; Ballantyne, C.M.; Folsom, A.R.; Pankow, J.S.; Selvin, E. Six-Year Change in High-Sensitivity C-Reactive Protein and Risk of Diabetes, Cardiovascular Disease, and Mortality. Am Heart J 2015, 170, 380–389, doi:10.1016/j.ahj.2015.04.017.
  4. Wang, X.; Bao, W.; Liu, J.; Ouyang, Y.-Y.; Wang, D.; Rong, S.; Xiao, X.; Shan, Z.-L.; Zhang, Y.; Yao, P.; et al. Inflammatory Markers and Risk of Type 2 Diabetes: A Systematic Review and Meta-Analysis. Diabetes Care 2013, 36, 166–175, doi:10.2337/dc12-0702.
  5. Liu, C.; Feng, X.; Li, Q.; Wang, Y.; Li, Q.; Hua, M. Adiponectin, TNF-α and Inflammatory Cytokines and Risk of Type 2 Diabetes: A Systematic Review and Meta-Analysis. Cytokine 2016, 86, 100–109, doi:10.1016/j.cyto.2016.06.028.
  6. Wang, Y.I.; Bettaieb, A.; Sun, C.; DeVerse, J.S.; Radecke, C.E.; Mathew, S.; Edwards, C.M.; Haj, F.G.; Passerini, A.G.; Simon, S.I. Triglyceride-Rich Lipoprotein Modulates Endothelial Vascular Cell Adhesion Molecule (VCAM)-1 Expression via Differential Regulation of Endoplasmic Reticulum Stress. PLoS One 2013, 8, e78322, doi:10.1371/journal.pone.0078322.
  7. Aburawi, E.H.; AlKaabi, J.; Zoubeidi, T.; Shehab, A.; Lessan, N.; Al Essa, A.; Yasin, J.; Saadi, H.; Souid, A.-K. Subclinical Inflammation and Endothelial Dysfunction in Young Patients with Diabetes: A Study from United Arab Emirates. PLoS One 2016, 11, e0159808, doi:10.1371/journal.pone.0159808.
  8. de Lima, E.P.; Moretti, R.C.; Torres Pomini, K.; Laurindo, L.F.; Sloan, K.P.; Sloan, L.A.; Castro, M.V.M. de; Baldi, E.; Ferraz, B.F.R.; de Souza Bastos Mazuqueli Pereira, E.; et al. Glycolipid Metabolic Disorders, Metainflammation, Oxidative Stress, and Cardiovascular Diseases: Unraveling Pathways. Biology (Basel) 2024, 13, 519, doi:10.3390/biology13070519.
  9. Giulietti, A.; van Etten, E.; Overbergh, L.; Stoffels, K.; Bouillon, R.; Mathieu, C. Monocytes from Type 2 Diabetic Patients Have a Pro-Inflammatory Profile. 1,25-Dihydroxyvitamin D(3) Works as Anti-Inflammatory. Diabetes Res Clin Pract 2007, 77, 47–57, doi:10.1016/j.diabres.2006.10.007.
  10. Larionova, I.; Patysheva, M.; Iamshchikov, P.; Kazakova, E.; Kazakova, A.; Rakina, M.; Grigoryeva, E.; Tarasova, A.; Afanasiev, S.; Bezgodova, N.; et al. PFKFB3 Overexpression in Monocytes of Patients with Colon but Not Rectal Cancer Programs Pro-Tumor Macrophages and Is Indicative for Higher Risk of Tumor Relapse. Front Immunol 2022, 13, 1080501, doi:10.3389/fimmu.2022.1080501.
  11. Cassetta, L.; Fragkogianni, S.; Sims, A.H.; Swierczak, A.; Forrester, L.M.; Zhang, H.; Soong, D.Y.H.; Cotechini, T.; Anur, P.; Lin, E.Y.; et al. Human Tumor-Associated Macrophage and Monocyte Transcriptional Landscapes Reveal Cancer-Specific Reprogramming, Biomarkers, and Therapeutic Targets. Cancer Cell 2019, 35, 588-602.e10, doi:10.1016/j.ccell.2019.02.009.
  12. Qi, J.; Sun, H.; Zhang, Y.; Wang, Z.; Xun, Z.; Li, Z.; Ding, X.; Bao, R.; Hong, L.; Jia, W.; et al. Single-Cell and Spatial Analysis Reveal Interaction of FAP+ Fibroblasts and SPP1+ Macrophages in Colorectal Cancer. Nat Commun 2022, 13, 1742, doi:10.1038/s41467-022-29366-6.
  13. Vale, C.; Godolphin, P.J.; Fisher, D.; Horby, P.W.; Kosiborod, M.N.; Hochman, J.S.; Webster, K.; Higgins, J.P.T.; Althouse, A.D.; Berwanger, O.; et al. Sodium-Glucose Co-Transporter-2 Inhibitors for Hospitalised Patients with COVID-19: A Prospective Meta-Analysis of Randomised Trials. Lancet Diabetes Endocrinol 2024, 12, 735–747, doi:10.1016/S2213-8587(24)00219-5.
  14. Hu, Z.; Liao, Y.; Wang, J.; Wen, X.; Shu, L. Potential Impacts of Diabetes Mellitus and Anti-Diabetes Agents on Expressions of Sodium-Glucose Transporters (SGLTs) in Mice. Endocrine 2021, 74, 571–581, doi:10.1007/s12020-021-02818-7.

Comment: I suggest discussing the mechanisms involved in the intracellular signaling induced by SGLT2i. Are there SGLT2 transporters in macrophages, muscle cells, endothelial cells, among others? Is the transport coupled with scaffold proteins responsible of the intracellular signaling? What is the opinion of the expert?

Answer: Figure 3 illustrates the molecular mechanisms of SGLT2 inhibitors in modulating anti-inflammatory signaling pathways in macrophages. For the first time we provided an overview of SGLT2 inhibitors, including empagliflozin, dapagliflozin, and canagliflozin, on key macrophage anti-inflammatory signaling pathways. SGLT2 inhibitors interact with various components of the inflammatory signaling cascade, including TLR2/TLR4, JAK2/STAT1, and the NLRP3 inflammasome, suppressing activation of NF-κB and PI3K/Akt signaling pathways while activating STAT3. For the first time we provide the summary that these interactions lead to alterations in the expression of pro-inflammatory cytokines (e.g., TNF-α, IL-1β, IL-6, IFN-β), inducible nitric oxide synthase (iNOS), and cyclooxygenase-2 (COX-2), which contribute to the anti-inflammatory response and suppression of foam cell formation.

Additionally, it was summarized and illustrated  for the first time that SGLT2 inhibitors promote the formation of foam cells, which are involved in the development of atherosclerosis. Figure 4 for the first time provides the summary of cellular effects of SGLT2 inhibitors reducing inflammation by affecting cross-talk between macrophage and other cell types. The figure illustrates the cellular targets and mechanisms influenced by SGLT2 inhibitors across different cell types, including endothelial cells, smooth muscle cells, macrophages, T-cells, and renal epithelial cells. This summary emphasizes the potential of SGLT2 inhibitors to influence not only glucose metabolism but also vascular and immune cell functions, contributing to their therapeutic benefits in cardiovascular and renal protection. The mechanism of SGLT2 transporters function in different cell types are summarized in new Table 3 and corresponding text is added to the chapter 6.3. “SGLT2 control metabolic inflammation in multiple cell types”.

Round 2

Reviewer 2 Report

Comments and Suggestions for Authors

Authors describe the anti-inflammatory effect of SGLT2 inhibitors in diabetic patients and animal model focusing on macrophages.

Authors performed satisfactory changes in the manuscript considering my observations. However, it is necessary to modify the title because Authors did not describe nonglycemic “effects” of SGLT2i, they focus only to anti-inflammatory effect. I suggest “Anti-inflammatory effect of SGLT2 inhibitors: focus on macrophages”.

The manuscript have mistakes of typing.

Authors should consult mainly original articles when they writing revision articles on future occasions.

Comments on the Quality of English Language

The English could be improved.

Author Response

We thank the reviewer for excellent suggestion about the title. The title is changed to: “Anti-inflammatory Effect of SGLT2 Inhibitors: Focus on Macrophages”.

We also performed English editing  in the revised version.